# Utilization of Formal and Informal Care by Community-Living People with Dementia: A Comparative Study between Sweden and Italy

**DOI:** 10.3390/ijerph15122679

**Published:** 2018-11-28

**Authors:** Carlos Chiatti, Danae Rodríguez Gatta, Agneta Malmgren Fänge, Valerio Mattia Scandali, Filippo Masera, Connie Lethin

**Affiliations:** 1Department of Health Sciences, Faculty of medicine, Lund University, Box 157, SE-221 00 Lund, Sweden; danaerodriguezg@gmail.com (D.R.G.); agneta.malmgren_fange@med.lu.se (A.M.F.); 2Department of Biomedical Sciences, Polytechnic University of Marche, 60030 Ancona, Italy; valeriomattiascandali@gmail.com; 3Department of Health Care Planning, Regional Health Agency of Marche Region, 60015 Ancona, Italy; filippo.masera@regione.marche.it; 4Department of Clinical Sciences, Clinical Memory Research Unit, Faculty of Medicine, Lund University, SE-214 28 Malmö, Sweden; connie.lethin@med.lu.se

**Keywords:** dementia, health services, resource utilization, dementia care, informal care, formal care, cognitive disorder

## Abstract

*Background:* Dementia is a public health priority with a dramatic social and economic impact on people with dementia (PwD), their caregivers and societies. The aim of this study was to contribute to the knowledge on how utilization of formal and informal care varies between Sweden and Italy. *Methods:* Data were retrieved from two trials: TECH@HOME (Sweden) and UP-TECH (Italy). The sample consisted of 89 Swedish and 317 Italian dyads (PwD and caregivers). Using bivariate analysis, we compared demographic characteristics and informal resource utilization. Multiple linear regression was performed to analyze factors associated with time spent on care by the informal caregivers. *Results:* Swedish participants utilized more frequently health care and social services. Informal caregivers in Italy spent more time in caregiving than the Swedish ones (6.3 and 3.7 h per day, respectively). Factors associated with an increased time were country of origin, PwD level of dependency, living situation, use of formal care services and occupation. *Conclusions:* Care and service utilization significantly varies between Sweden and Italy. The level of formal care support received by the caregivers has a significant impact on time spent on informal care. Knowledge on the factors triggering formal care resources utilization by PwD and their caregivers might further support care services planning and delivery across different countries.

## 1. Introduction

Dementia is a disease currently affecting around 50 million people worldwide, corresponding to about 5% of the older population worldwide [1]. It is a public health priority for which most of the modern welfare states are not fully prepared [2]. Dementia onset is associated with age and low education in early life [1,3]. Other causal determinants include hypertension at midlife, smoking and diabetes, which are often related to lifestyle factors [3]. People with dementia face a disabling condition and gradually reach the need for extensive support by informal and formal care in activities of daily living (ADL), e.g., showering, dressing and managing finances [3], restricting their participation in the society leading to adverse events such as institutionalization and inappropriate use of health care and social services (care and services) [3]. The disease has no cure and affects considerably also the life of families and significant others, leading to an overall negative economic and health impact for the society [4]. Indeed, empirical data show that everywhere in Europe families and other informal caregivers are those who provide the bulk of care to their dependent older relatives, thus facing the most significant share of the disease burden [5].

The World Health Organization (WHO) strongly recommends addressing this challenge by creating specific dementia care plans in each country and accordingly reallocating resources [1]. In line with these recommendations, high-income countries in recent years endeavored to achieve effective health prevention, timely diagnosis and an overall increase of the availability of health care and social services (care and services) [6]. Access to long-term care services is particularly relevant due to the dementia characteristic trajectories during which people might require a mix of care and services to ensure independent living and quality of life [7]. The care is mainly related to the timely diagnosis and symptom management and services are mainly focusing on supporting independence in ADL. The availability of care and services may differ significantly across countries, e.g., in terms of type and amount of services provided and the way they are delivered [8]. Recent evidence suggests that this intervention area could include interventions to improve lifestyle, e.g., in terms of physical activity and nutrition, as this might prevent complications and even slow down the progression of the disease [9]. As the majority of people with dementia live at home, due to the deinstitutionalization and ageing-in-place policies, informal caregivers, mainly family members, make up the cornerstone of the whole care system [3,4]. However, as a consequence of their difficult roles, they experience a high level of caregiver burden which put them in a condition of psychological, physical and financial strain [1,3], being at risk of declining health and increased care needs. As the on-going trends suggest a future increase of the prevalence of people with dementia worldwide, there is an urgent need to improve the effectiveness and efficiency of the dementia care systems in all countries. It is especially relevant to achieve an appropriate balance between the formal and informal care contribution to guarantee care situations respectful of the quality of life and social rights of both people with dementia and their caregivers.

In this respect, several studies have analyzed the resource utilization in dementia care [10] across different geographical and cultural contexts, also using a comparative approach [11]. These studies have revealed how widely the use of care and services can vary across countries, and how different the conditions of the informal caregivers can be. Nonetheless, many studies have adopted a pure cost-analysis approach, providing mostly data on the global financial consequences of the disease (e.g., in terms of total public or private healthcare expenditure), without giving details on the characteristics and intensity of the services driving the costs and their relation with the informal care network. Comparative analysis in this area would be particularly interesting within the European Union context, and the similarities and differences in how the dementia care systems are organized [12]. Sweden and Italy for instance represent different welfare systems and cultures (Nordic vs. Mediterranean), facing similar challenges in organizing dementia care and support for informal caregivers. Both Sweden and Italy are high income countries with similar life expectancy at birth, estimated at 83 and 82 years for Italy and Sweden, respectively [13,14]. Sweden is a Nordic welfare state with a long tradition of service provision to support people with dementia and their caregivers [15,16,17]. The population size is around 10 million inhabitants, representing one of the oldest populations in the world [18,19], out of which approximately 160,000 people have a dementia diagnosis [20]. Dementia is one of the leading causes of mortality in Sweden [14]. The majority of people with dementia live at home with support from informal caregivers and have a wide range of available care and services [20]. The care of people with dementia in Sweden is a shared responsibility between the counties and the municipalities. The county councils are responsible for healthcare according to the Health and Medical Services Act [21]. The municipalities are responsible for the elderly citizens, 65 years or older, according to the Social Services Act [22]. All care is financed through taxes. National guidelines for dementia care and the provision of care and services have been developed to ensure that the care provided by county councils and municipalities are equal independently of where one lives [21]. Italy has a population of approximately 60 million people and the second highest proportion of older people (65 years old or more) in Europe and about 1 million people with dementia [23,24]. The healthcare system is regionally decentralized, and Italy was the first country to introduce nationwide specialized memory clinics for dementia care [16,25]. Here, while home help is provided by the municipalities, nursing home care is the responsibility of the Local Health District and largely funded by the National Health Fund. However, significant disparities exist among Italian regions regarding the availability of these services [25]. Most often, families opt for privately paid care workers due to the scarce services available for dementia care, and thus they face a significant impact on household budgets due to out-of-pocket expenditures [23,26]. This in part urged the creation of a national plan that aims to promote and improve interventions for people with dementia and their families [25].

In terms of resources allocated to healthcare, the total health expenditure in Italy reaches about 9% and in Sweden about 11% of GDP, while out-of-pocket expenditure of the Italian population accounts for 23% of their current health expenditure, compared to 15% for Sweden [27]. Due to the different allocations of resources within the healthcare budget and the different levels of resources available in the area of care and services, it is estimated that, on average, a Swedish citizen has at least three times more funding available for long-term care compared to an Italian citizen [15].

This study was based on the experience of two recent research projects, namely TECH@HOME and UP-TECH, which have investigated, among other outcomes, the economic impact of dementia in Sweden and Italy. The similarities of the study designs and samples give the opportunity to investigate two different care systems responding to similar challenges concerning welfare systems sustainability. Such comparison might contribute to a better understanding of the current dementia care and service systems in Europe, but also to hypothesize how differences in the levels of formal care provision might impact on the dynamics of informal care levels, and vice versa [28]. The overarching aim of this study was thus to contribute to the knowledge gap on utilization of formal and informal dementia care across European welfare states and on how such differences might impact on the daily arrangements of people with dementia and their caregivers. Specific aims of the study were to evaluate: (a) the characteristics of people with dementia and their caregivers in Sweden and Italy; (b) the level of resources utilization in the two countries; and (c) the factors associated with the time spent in caregiving by the informal caregivers of the people with dementia.

## 2. Materials and Methods

### 2.1. Study Design and Sample

This study had a comparative cross-sectional design, drawing on data from two studies. TECH@HOME [29] is an ongoing prospective study conducted in southern Sweden between 2016 and 2019, focusing aspects of physical and mental health, health related quality of life and use of health care and social services among PwD and their informal caregivers, after installation of a sensors-based monitoring system in their homes. The UP-TECH (Chiatti et al.) [30] project is a multi-component, randomized controlled trial recently conducted in the Italian Marche region during 2014–2015. The main objective was to reduce caregiver burden of family caregivers of patients with Alzheimer’s disease (AD) and to maintain patients with AD at home for as long as possible, through the use of new technologies and of case-management approaches. Inclusion criteria for the two studies were very close, although not completely overlapping. In TECH@HOME, the inclusion criteria were: a diagnosis of major neurocognitive disorders with a mild to moderate severity; a Mini Mental State Examination (MMSE-SR) between 14 and 24 [31]; a Global Deterioration Scale (GDS) score between 1 and 5; living at home; being able to understand and communicate in Swedish; and having an informal caregiver [29]. Dyads were enrolled through the primary health care centers and the municipalities. Inclusion criteria of UP-TECH were [30]: Alzheimer’s diagnosis at an intermediate stage; and MMSE between 10 and 20. Potential participants were recruited at the Alzheimer Evaluation Units [30]. To improve the comparability of the two samples, we selected in this study only people with dementia with a MMSE between 14 and 24 and their caregivers. The final sample resulted in 89 Swedish and 317 Italian dyads.

### 2.2. Data and Measures

Basic socio-demographic data such as gender, age, marital status and living situation of the person with dementia were available for both the TECH@HOME and the UP-TECH samples, as well as data on the age, gender, occupation, education and caring situation of the caregivers. In addition, the questionnaires administered by the research nurses in both studies included reliable and valid instruments to address several clinical (related to diseases and other clinical conditions) and functional dimensions (related to the individual cognitive and physical functioning, e.g., ADL, IADL and MMSE). The rationale for also evaluating these dimensions in our study is that literature has vastly confirmed the existence of multiple explanatory factors behind the use of health and social care resources. The Andersen–Newman healthcare utilization model [32], for instance, suggests that use of resources could depend on the actual needs of a person as well as on the socio-environmental characteristics of the individual and his/her family. The procedures used for data collection are thoroughly described in details elsewhere [29,30]. In both studies, information on cognitive function, as measured by the MMSE [31], was available. The MMSE is a 30-point questionnaire extensively used in clinical and research settings to measure cognitive impairment. A score greater or equal to 24 points indicates intact functioning, while below this threshold can indicate severe (≤9), moderate (10–18) or mild (19–23) cognitive impairment. Data on ADL were collected by means of the Interrai ADL Hierarchy Scale [33]. This scale includes four items rating the functional status in relation to self-performance (i.e., personal hygiene, toilet use, locomotion, and eating), which are summarized in a hierarchical scale that ranges from 0 (no impairment) to 6 (totally dependent). Dependence in IADL were assessed using the IADL Involvement Scale, which is based on seven IADL-related items, summarized in a scale that ranges from 0 to 48, with higher scores indicating greater dependency [34]. The Hospital Anxiety and Depression scale (HADS) was used in both studies to assess caregiver’s level of psychological health [35]. HADS includes 14 items; seven items related to anxiety and seven items related to depression. Each item is assessed on a Likert scale that ranges from 0 to 3 and the overall scores range from 0 to 21 in both anxiety and depression where 0–7 is consistent with absence of the conditions [35].

### 2.3. Use of Resources

The instrument used for data collection in TECH@HOME was the Resource Utilization in Dementia (RUD) [36]. This instrument has been extensively used especially for cost analysis and has a widespread use in global settings. The RUD instrument collects information regarding the level of resource utilization by frequency in hours and days, from both the person with dementia and the caregiver, together with other demographic and health status information. The UP-TECH study used an ad-hoc developed Resource Utilization Form, which had several sections overlapping with those included in the RUD, therefore providing comparable data for the analysis [30].

Some variables have been be recoded to obtain identical unit of measurements: for example, outpatient visits in Italy were measured using a six-month timeframe, while in the Swedish questionnaire they were measured using a 30-day period. Time spent in care activities by informal caregivers in TECH@HOME has been assessed using the specific section of the RUD instrument, while in UP-TECH one item assessed the hours spent in caregiving activities by the primary caregiver during the day. A second item retrieved the time spent by other secondary caregivers.

### 2.4. Data Analysis

In a first step, samples were compared according to socio-demographic characteristics, clinical and functional measures. Differences were investigated using bivariate analysis. Statistical significance for categorical variables was assessed using the Chi-squared or the Fisher’s exact test. For continuous variables, independent T-test and Mann–Whitney test were used to compare means depending on the distribution of the investigated variable (normal vs. non-normal distribution). In a second step, differences in the level of use of formal care services and in the amount of informal care provided were evaluated following a similar statistical procedure. Finally, multiple linear regression models were built to evaluate factors associated with informal caregiving time. Potential factors were tested in the model if they had a *p*-value ≤ 0.25 following the results of the bivariate analysis and inserted in the model using a step-forward. Different models were tried, and variables were dropped from the model when they decreased the Adjusted R-squared, had no significant results or had no impact in the rest of the parameters. The coefficients described in the tables are based on 1000 bootstrap samples. Statistical significance was considered with a *p*-value ≤ 0.05. The statistical software package used was STATA (Statacorp, College Station, TX, USA) [37].

## 3. Results

### 3.1. Demographic Characteristics of Persons with Dementia and Informal Caregivers

Mean age among study participants with dementia was higher in Italy compared to Sweden (81.5 vs. 78.4, *p* < 0.001). Nonetheless, the largest age group in both countries was that of people between 80 and 89 years old (Table 1). Most people in both countries were female and living with their spouses. However, people with dementia in Sweden tend to live alone more frequently (47.2% in Sweden vs. 15.1% in Italy, *p* < 0.001). In Italy, it was frequent that a person with dementia lives with an adult child, which was really uncommon in the Swedish context (only in 1.1% of the cases). From a clinical and functional point of view, the Italian participants seem characterized by an overall worse condition. Cognitive function was slightly lower in Italy where the mean MMSE value is 17.8 (SD = 2.3) (vs. 19 ± 3.7 in Sweden). Additionally, the Italian group was characterized by a higher number of comorbidities as well as by higher level of dependency measured using the ADL and IADL scale.

The majority of informal caregivers were women, although in Italy women were more represented than in the Swedish context (68.8% vs. 52.3%). Caregivers in Italy were younger, and more often an adult child living with the person with dementia. In Sweden, the informal caregivers more often were spouses, which tend to be more frequently engaged in a working occupation.

### 3.2. Resource Utilization by Persons with Dementia and Their Caregivers

Results showed that the Swedish participants more frequently used healthcare and social service than the Italian sample, except for emergency care (Table 2). However, hospital admissions were more frequent in Sweden compared to Italy (5.6 vs. 1.6, *p* < 0.001). The most frequently used services in Sweden were day care centers (used by 39.8% of the sample), social services (65.9%) and home care (38.6%). In Italy, the main support for the people with dementia and their caregiver was the use of privately paid home help services (used by 28.7% of the sample).

### 3.3. Time Spent in Informal Caregiving

Italian informal caregivers spent significantly more time in all caring activities than Swedish ones and had a mean of 6.27 h per day (vs. 3.5 h per day spent by Swedish caregivers) (Table 3). In Italy, the number of hours spent in caregiving activities is higher when the person with dementia is a male (7.47 vs. 5.76, *p* = 0.006) and the caregiver is a female (6.49 vs. 5.81, *p* value is not significant). In both countries, the hours of caregiving provided increase with the level of dependency in ADL and IADL. Living with the person with dementia is not associated with a higher amount of care provided either in Sweden or in Italy.

The results of the multiple regression model showed that higher IADL dependence, working status of the caregiver, use of formal home care, and the country of origin were factors associated with the number of hours of informal care provided (Table 4). The lower the IADL dependence, the higher the amount of care provided (+4.6 h per day, among people with dementia with an IADL score between 44 and 48 (*p* < 0.001). Conversely, working carers provided 4.2 h less care compared to the non-working ones (*p* < 0.001). Compared to Italian couples not using any type of formal care support, those Italians using private care help provided - 1.8 h care/day. Being Swedish, even if not using home help from the municipality, was associated with a reduction of the number of daily hours of care equal to 2.6 (*p* < 0.001).

## 4. Discussion

### 4.1. Differences in Socio-Economic Conditions of the Dyads in the Two Countries

Findings of our study showed mainly “women caring for women”, as in both countries they were the majority among caregivers and people with dementia. This trend has been seen in many other studies where female gender is highly represented [38,39,40]. However, the Swedish commitment with gender equality was revealed in our findings. In line with other studies focusing on long-term care in a Swedish context, only slight differences were seen between the proportion of male and female informal caregivers [17,18], differently from what was observed in Italy, where the ratio of female to male informal caregiver is about 7 to 3.

Diverse caring dynamics were seen in each country. In Italy, informal caregivers tend to live significantly more with the person with dementia than in the Swedish sample. Some European studies have shown that living with the PwD indicates higher time spent in caregiving [41,42,43] and higher risk of caregiver burden [41,44]. These precedents coincide with our findings. Italian caregivers had a higher time spent in caregiving activities and had a worse psychological health than the Swedish sample. Interestingly, when adding IADL dependence level to the model, the relation between the variable “living with the PwD” and “hours of informal care provided” changes. At a bivariate level, living together the PwD was associated to a higher number of hours of care provided. However, our regression suggests that this might be the result of the interaction between the living status and the IADL dependence level of the PwD. In other terms, caregivers caring for PwD with higher IADL impairment are more frequently living with them, compared with those caring for PwD with less level of dependency. This suggests that the choice of co-residence might be influenced by the level of dependency of the person with dementia.

In this respect, it is interesting to observe that Italian persons with dementia were more frequently living with their adult children. This difference may be explained due to a still high prevalence of intergenerational households in Italy and prevalent family-based care of the country [16,38,45]. This possibly reflects a history of a strong cultural norm of caring responsibilities, in spite of its current changing trend inside families [45]. It could also reflect a cost saving strategy. Households under economic strain and the financial situation of a country contribute in the decision of co-residence [46]. This could be intentionally opted for or more a response to the increasing uncertainties in the globalized world [46]. Conversely, in Sweden, people with dementia were more frequently living alone, probably due to the widespread support from the state, which could offset economic burden and allow them to sustain living on their own [46,47].

Surprisingly, even though Swedish caregivers represented an older age group, our findings showed that they were more actively working than the Italian caregivers. The extensive utilization of dementia care services might allow them to remain at the workplace and consequently spend fewer hours per day in caregiving. Conversely, the Italian sample was less actively working despite having more caregivers in a working age. As previously depicted in other studies, they might have to reduce their working hours or to stop working to care for the person with dementia [23,44]. In line with these results, our regression model indicated an association between caregivers actively working and less time spent in informal caregiving activities.

### 4.2. Differences in Informal Caregiving Intensity

The time spent in informal care was regarded as an important part of the analysis. In a pan-European study, Bremer and Cabrera [48] considered that an average of 4–8 h per day spent in caring for ADLs by informal caregivers constitutes a medium level of intensity of care. Similarly, Ory and Hoffman [49] utilized an intensity of care index where about 3 h per day is considered an intermediate level of care. According to these definitions, we could conclude that both countries significantly relied on informal care. The average time spent in caring activities was 3.9 h per day in the Swedish sample and 6.3 h per day in the Italian one. The age of the PwD was associated with informal caregiving time, and this is consistent with previous studies conducted in a European context [41,50]. Additionally, studies have reported an association between high severity of disease and worse health state of the informal caregiver [42]. Italian people with dementia had a higher level of severity of the disease and this could explain the higher proportion of psychological distress in their informal caregivers.

Interestingly, our regression showed that being from Sweden is associated with fewer mean hours spent in caregiving activities in comparison to the Italian sample. The impact of the country of origin on informal care might reflect differences in health care systems, and socioeconomic and cultural factors [40,51]. In our study, we found higher utilization of health and social care services in the Swedish sample, but lower levels of informal care provisions. These differences might be rooted in several explanatory factors. Firstly, Italy and Sweden represent two different types of welfare states, as suggested by Esping-Andersen work, in which different levels of complementarity between informal and formal care can be found [12,15]. Welfare states can be categorized as liberal, conservative and social democratic and differences among these typologies reflect their “(...) political ideologies with regard to stratification, de-commodification and the public-private mix of welfare” [12]. A social democratic model, such as the Swedish one, represents a state that ensures equal health and social service provision and funding for all citizens through a tax-based system [38]. In a conservative model, such as the Italian one, the state is partially responsible for service provision and funding through social insurance schemes, therefore relying on a strong family-based care. Raggi and Leonardi’s work [52] is in line with this interpretation, as it identifies a north–south European gradient in care, in which Northern European countries are characterized by universal social policies, state support for families and a large public sector and Southern European one by a mix of universal private services and benefits together with a fragmented system between health and social care services. Our findings indeed show that the Italian sample had a comparatively low utilization of services but strongly relied either in another informal caregiver or in paid care workers. Secondly, cultural norms could be considered another cause of divergence between countries. As Brandt and Haberkern pointed out [38], intergenerational help is subject to cultural norms, and according to these norms the state would have an effect in “crowding in” (more family support) or “crowding out” (less family support). For instance, in the Swedish context, intergenerational households are comparatively scarcer and adult children are less frequently caregivers than in the Italian context. The state here supports the families through services, thus creating a “crowding out” effect. Conversely, in Italy, more adult children are informal caregivers and the state plays a residual role, thus creating a “crowding in” effect [38,53]. Due to lack of services, Italian families might have slim alternatives of care and internalized caring responsibilities and this could also reflect their increased level of burden seen in our findings. Thirdly, help seeking behaviors constitute another potential source of explanation of the differences seen in resource utilization between the two countries in our study. According to Alzheimer’s Disease International (ADI) “concerns regarding stigma may be one factor deterring or delaying help seeking” [6]. This could be explained by the lack of information and awareness of services available together with a need of improvement in tailored formal care delivery [54]. In the case of the Italian informal caregivers, a perception of self-sufficiency and a reliance on internal family support might hinder the demand for other alternatives of care [48]. Lastly, availability and accessibility of services influences the level of resource utilization. The dementia care pathway should be addressed through a continuum between health and social care services across “(...) the various stages of the disease as a seamless process, as needs for both types of care evolve” [8]. Fragmentation between these could potentially hinder utilization of resources. In this respect, formal services in Italy are often described as scarce and fragmented, hence this could explain the low resource utilization seen in our findings [15,55]. In the Swedish context, formal services seem to be highly available but its utilization is low, therefore an alignment with needs is necessary [56].

### 4.3. Strengths and Limitations

Among the strengths of our study is the possibility to compare countries due to a highly comparable sample of participants and research designs. Eligibility criteria and questionnaire structures were similar across the two clinical trials. However, cross-country comparison becomes challenging in presence of strong differences in how the services themselves are organized across welfare states. For instance, each country offers different types of social care services, in terms of contents and intensity, therefore their comparison should be taken with caution. Similarly, the variable that we have created for formal home help pointed at two different services in the two countries. Swedish formal caregivers have basic healthcare training while in Italy they are in most cases people without a formal qualification with a migration background [57]. Furthermore, the data available for analysis represent samples from specific regions of each country and significant variation could be found within and across them. Thus, the generalizability of the findings at country-level could be limited. In addition, given the cross-sectional design of our primary studies, other important factors that could be involved in determining the dynamics of care might have overlooked, i.e., generating a so-called ecological bias. Finally, informal care could entail support to several different activities, e.g., ADL, IADL or supervision [41]. Detailed comparisons of informal care time could not be performed in our study, since more details information where not included in the Italian data.

## 5. Conclusions

Our paper suggests that differences in the organization of health and social care systems across welfare states have broad impacts. They influence the time spent in informal care, the amount of reliance on family support, gender roles, working status and levels of psychological burden among populations. Our findings may facilitate the understanding of dementia care system variation across countries, thus having an impact for both policy and care management. This knowledge is also valuable for professionals and students in educational training, preparing for careers in the health care and social service area. Differences between Sweden and Italy are quite significant and rooted in diverse explanations. Balance between informal and formal care is essential for welfare states sustainability and similar resource utilization dynamics can provide crucial information regarding the informal–formal care mix [54]. In the future, further investigation could be made including analysis of the unmet needs. In this respect, further qualitative research could be necessary to explore the experiences of people during their debate on seeking support services or not in the dementia care pathway.

## Figures and Tables

**Table 1 ijerph-15-02679-t001:** Baseline characteristics of persons with dementia and informal caregivers.

	Swedish Sample (n = 89)		Italian Sample (n = 317)		
	%	Mean (SD)	%	Mean (SD)	*p **
**Persons with dementia**					
**Age in years, (%)**	-	78.4 (7.8)	-	81.5 (5.8)	0.001
Early old age (<69)	11.2	-	3.5	-	0.009
Middle old age (70–79)	39.3	-	32.5	-	
Later old age (80–89)	44.9	-	59.3	-	
Very old age (>90)	4.5	-	4.7	-	
**Gender**					
Men	27.0	-	30.3	-	0.544
Women	73.0	-	69.7	-	
**Living situation**					
Living alone	47.2	-	15.1	-	<0.001
Husband/wife	49.4	-	50.2	-	
Child	1.1	-	21.5	-	
Other	2.2	-	13.2	-	
**Comorbidity, n of diseases**	-	1.7 (1.1)	-	2.2 (1.4)	0.001
**Cognitive function, MMSE score (0–30) ^a^**	-	19 (3.7)	-	17.8 (2.3)	<0.001
Mild (≥20)	53.0	-	54 (17.0)	-	<0.001
Moderate (10–19)	47.0	-	263 (83.0)	-	
**ADL scale (0–6) ^b^**	-	0.2 (0.5)	-	1.4 (1.5)	<0.001
**IADL scale (0–48) ^c^**	-	25.2 (12.4)	-	33.3 (14.2)	<0.001
**Informal caregivers**					
**Age-groups (%) ^d^**					
<39–54	29.6	-	39.1	-	0.051
55–69	39.8	-	31.2	-	
>70	30.7	-	29.7	-	
**Gender ^d^**					
Men	47.7	-	31.2	-	0.004
Women	52.3	-	68.8	-	
**Marital status ^d^**					
Married	74.2	-	79.8	-	0.031
Divorced/widowed	2.2	-	7.9	-	
Never married	22.5	-	12.3	-	
**Occupation ^e^**					
Employee/self–employed	50.0	-	46.1	-	0.001
Job seeking	4.7	-	3.8	-	
Retired	36.0	-	36.0	-	
Sickness/activity allowance/sick leave ^f^	5.9	-	2.2	-	
Other	3.5	-	12.0	-	
**Level of education ^d^**					
Elementary school	14.8	-	7.6	-	0.074
Gymnasium	36.4	-	33.8	-	
University	48.9	-	58.7	-	
**Relationship to the person with dementia ^d^**					
Wife/Husband	43.2	-	30.9	-	0.038
Child	50.0	-	54.6	-	
Other	6.8	-	14.5	-	
**Living with the person with dementia**					
Yes	45.5	-	63.7	-	0.002

Data source for Sweden: TECH@HOME questionnaire. Data source for Italy: UP-TECH questionnaire; ADL, activities of daily living; IADL, instrumental activities of daily living; n, number of observations; MMSE, mini mental state examination; *p* value of significance; SD, standard deviation. ^a^ missing values, n = 6; ^b^ missing values, n = 2; ^c^ missing values, n = 4; ^d^ missing values, n = 1; ^e^ missing values, n = 3; ^f^ long term sick leave. * *p* < 0.05 was regarded as significant; significant *p*-values are underlined. Underlined values indicate positive results, e.g., 0–30.

**Table 2 ijerph-15-02679-t002:** Resource utilization the last 30 days of healthcare and social services by persons with dementia.

	Swedish Sample (n = 89)		Italian Sample (n = 317)		
	%	Mean (SD)	%	Mean (SD)	*p* *
**Hospital admission (%)**					
Yes	5.6	-	1.6	-	0.045
Number of admissions	-	1.6 (1.3)	-	-	
**Emergency ward admission ^b^**					
Yes	4.6	-	6.3	-	0.798
No	95.4	-	93.7	-	
Number of visits	-	1.3 (0.5)	-	1 (0)	0.477
**Outpatient care**					
Yes	38.2	-	9.1	-	<0.001
Number of visits	-	1.7 (1.3)	-	1 (0)	0.001
**Municipal (SWE) or District (ITA) nurse ^c^**					
Yes	12.6	-	2.8	-	0.001
Number of visits	-	3.36 (2.7)	-	3 (3.5)	0.664
**Day care centre**					
Yes	39.8	-	3.8	-	<0.001
Number of visits	-	11.4 (5.4)	-	1 (0)	<0.001
**Social care services ^c^**					
Yes	65.9	-	6.9	-	<0.001
**Home help (SWE) or Private Carer (ITA) ^b^**					
Yes	38.6	-	28.7	-	0.074

Data source for Sweden: TECH@HOME questionnaire. Data source for Italy: UP-TECH questionnaire. ITA, Italy; *p*, *p*-value; SD, standard deviation; SWE, Sweden. ^a^ Median and interquartile range is only provided in case the variable was non-normally distributed: ^b^ missing values, n = 1; ^c^ missing values, n = 2. * *p* < 0.05 was regarded as significant; significant *p*-values are underlined.

**Table 3 ijerph-15-02679-t003:** Time spent in informal caregiving and psychological health of informal caregivers.

	Swedish Sample (n = 89)			Italian sample (n = 317)		
	N (%)	Mean (SD)	*p **	N (%)	Mean (SD)	*p **
**Patients characteristics**						
Age in years			0.205			0.230
<69	10 (11.4)	3.22 (3.63)		11 (3.5)	3.96 (3.49)	
70–79	35 (39.8)	3.92 (3.27)		103 (32.5)	6.26 (6.98)	
80–89	39 (44.3)	3.74 (5.03)		188 (59.3)	6.23 (6.13)	
>90	4 (4.5)	1.80 (0.91)		15 (4.7)	8.64 (7.72)	
Gender			0.351			0.006
Male	24 (27.3)	3.60 (2.65)		96 (30.3)	7.47 (6.72)	
Female	64 (62.7)	3.69 (4.55)		221 (69.7)	5.76 (6.26)	
MMSE, score (0–30)			0.006			0.244
Mild (≥20)	43 (52.4)	2.98 (3.63)		54 (17.0)	6.11 (6.80)	
Moderate (10–19)	39 (47.6)	4.44 (4.55)		263 (83.0)	6.31 (6.38)	
ADL scale score (0–6)			0.304			0.002
0	74 (84.1)	3.24 (3.30)		125 (39.4)	4.73 (5.26)	
1	8 (9.1)	6.69 (7.49)		72 (22.7)	7.26 (7.13)	
2	4 (4.5)	5.59 (7.65)		61 (19.2)	7.26 (7.17)	
3 or more	2 (2.3)	3.50 (0.71)		59 (18.7)	7.32 (6.55)	
IADL scale score (0–48)			<0.001			<0.001
1–22	32 (38.1)	2.66 (3.70)		74 (23.3)	3.30 (3.48)	
23–35	34 (40.5)	3.31 (2.42)		69 (21.8)	6.79 (6.71)	
36–43	13 (15.5)	5.91 (4.35)		81 (25.6)	6.62 (6.68)	
44–48	5 (5.9)	9.35 (8.93)		93 (29.3)	7.97 (7.10)	
Formal Caregiver						
No	53 (60.9)	3.62 (3.82)	0.910	226 (71.3)	6.79 (6.93)	0.062
Yes	34 (39.1)	3.68 (4.62)		91 (28.7)	4.99 (4.81)	
**Caregivers characteristics**						
Gender			0.515			0.697
Male	42 (47.7)	3.76 (4.41)		99 (31.2)	5.81 (5.56)	
Female	46 (52.3)	3.58 (3.86)		218 (68.8)	6.49 (6.80)	
Status in employment			0.036			<0.001
Employed	43 (50.0)	3.28 (4.38)		146 (46.1)	3.35 (3.09)	
Not employed	43 (50.0)	3.74 (3.01)		171 (53.9)	8.77 (7.44)	
Living conditions			0.015			0.170
Living with the PwD	40 (45.5)	3.98 (3.18)		200 (63.7)	6.33 (6.22)	
Not living with the PwD	48 (54.5)	3.40 (4.77)		114 (36.3)	6.23 (6.91)	
HADS Score Depression (0–21)			0.017			
Normal (0–7)	75 (86.3)	2.95 (3.20)		169 (53.3)	5.95 (6.30)	0.218
Mild (8–10)	9 (10.3)	7.54 (5.61)		79 (24.9)	7.74 (7.65)	
Moderate (11–14)	3 (3.4)	8.78 (9.79)		47 (14.9)	5.55 (5.28)	
Severe (15–21)	0 (0.0)			22 (6.9)	5.10 (4.03)	
HADS Score Anxiety (0–21)			0.354			0.732
Normal (0–7)	83 (95.4)	3.58 (4.14)		202 (63.7)	6.55 (6.60)	
Mild (8–10)	4 (4.6)	4.65 (3.79)		57 (18.0)	5.68 (6.43)	
Moderate (11–14)	0 (0.0)			42 (13.3)	5.30 (5.00)	
Severe (15–21)	0 (0.0)			16 (5.0)	7.56 (7.79)	
Total sample	86 (100)	3.50 (3.74)	n.a.	317	6.27 (6.44)	n.a.

Data source for Sweden: TECH@HOME questionnaire. Data source for Italy: UP-TECH questionnaire. HADS, hospital anxiety and depression scale; *p* = *p*-value; n.a, not applicable; *p*, *p*-value; PwD, person with dementia; MMSE, mini mental state examination; SD, standard deviation. * *p* < 0.05 was regarded as significant; significant *p*-values are underlined. Underlining of values indicates positive results, e.g., 0–30.

**Table 4 ijerph-15-02679-t004:** Factors associated with hours of informal caregiving.

Independent Variables	Hours of Informal Caregiving (n = 392)		
	B (SE)	*p*-Value *	95% CI
Constant	6.715 (2.03)	0.001	(2.74;10.686)
**Relation with PwD (Partner/Spouse vs. Other)**	0.461 (0.600)	0.443	(−0.716; 1.638)
**Informal caregiver working (Yes vs. No)**	−4.152 (0.553)	<0.001	(−5.237; −3.068)
**Interaction between Country # Utilization of formal caregiving** (Ref. Italy/not using formal caregiving)			
Italy/using formal caregiving	−1.874 (0.684)	0.006	(−3.215; −0.533)
Sweden/not using formal caregiving	−2.631 (0.707)	<0.001	(−4.017; −1.245)
Sweden/using formal caregiving	−1.078 (0.961)	0.262	(−2.963; 0.805)
**IADL scale (1–48)**			
23–36	2.794 (0.692)	<0.001	(1.436; 4.152)
37–43	3.273 (0.789)	<0.001	(1.725; 4.821)
44–48	4.655 (0.914)	<0.001	(2.863; 6.447)
**Living with the PwD (Yes vs. No)**	−0.001 (0.324)	0.995	(−0.638; 0.634)
**ADL scale (one point increase)**	−0.002 (0.178)	0.990	(−0.351; 0.346)
**MMSE score (one point increase)**	−0.061 (0.109)	0.572	(−0.276; 0.152)
R-squared	0.244		
Adjusted R-squared	0.222		

Data source: TECH@HOME and UP-TECH questionnaire. ADL, activities of daily living; B (SE), Beta coefficient (standard error); CI, confidence intervals; MMSE, mini mental state examination; PwD, person with dementia. IADL, instrumental activities of daily living. Country of origin: 1 = Sweden, 0 = Italy. Living with the person with dementia: 1 = no, 0 = yes. Utilization of formal caregiver: 1 = yes, 0 = no. Informal caregiver actively working: 1 = yes, 0 = no. * *p* < 0.05 was regarded as significant; significant *p*-values are underlined.

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
