# Peer review of "Utilization of Formal and Informal Care by Community-Living People with Dementia: A Comparative Study between Sweden and Italy"

_ijerph, 2018, doi:10.3390/ijerph15122679_

Round 1

Reviewer 1 Report

Dear authors, thank you for the submission of your article to this journal. Here they are my comments:

Introduction

The role of family caregivers for the provision of care to patients with Alzheimer should be discussed more.

Also, you need to describe more details of informal and home care to these patients in the context of Italy and Sweden.

Methods

The design of the current study as comparative cross-sectional study should be recognized.

Please provide some more details of the studies of TECH@HOME and The UP-TECH as the background of your current study.

The data collection tolls should be discussed one by one and related citations for their origination and copyright should be mentioned.

You need to introduce the data collection tools one by one and provide related details, content, as scoring, validity, reliability, analysis, interpretation etc.

The data collection tools (HADS, MMSE, ADLs) do not match with your study aims. You need to revise the study aim or introduce related data collection tools in line with your aim.

Results

Why you have included HADS and ADLS findings to other variables?

Could you use some figures to compare some important parts of your findings?

Discussion

This section is fine, but when you use other studies for comparisons, you need to described some details of them such as country and settings etc.

Conclusion

What are implications of your findings for Education, Policy-making and management?

Author Response

Dear reviewer,

Thank you for the response to our manuscript  “Utilization of formal and informal care by community-living people with dementia: a comparative study between Sweden and Italy”

We have carefully considered your comments and revised the manuscript. Attached you can find our point-by-point response.

Reviewer 2 Report

The authors present a comparative study between Sweden and Italy in the Utilization of formal and informal care by community-living people with dementia. The authors have retrieved their data from two trials: TECH@HOME (Sweden) 19 and UP-TECH (Italy).

The paper is well organized, the introduction is clear, the methodology although well analyzed is complicated per se.

The study is interesting and the results confirm in a scientific way some information that is already known i.e that in southern countries like Italy  there is prevalence of intergenerational households and prevalent family based care reflecting among others cultural differences and a cost saving strategy while Northern European countries as Sweden are characterized by universal social policies, state support for families and a large public sector.

The discussion is quite extensive like a minireview showing how complicated the issue is.    

 The study has many limitations that the autors state by themselves while in their conclusion they refer to the need of a qualitative research.   

Comment

 It would be useful if the authors could summarize the scores used in the 2 studies in a table with the appropriate comments.

Author Response

We thanks the reviewer for the positive comments. We enclose a response to the main point raised.

Round 2

Reviewer 1 Report

Nothing.